# LEARNING TO LINEARIZE DEEP NEURAL NETWORKS FOR SECURE AND EFFICIENT PRIVATE INFERENCE

**Souvik Kundu** [*]
Intel Labs, San Diego, USA
souvikk.kundu@intel.com

**Shunlin Lu, Yuke Zhang, Jacqueline T. Liu, & Peter A. Beerel**
Department of Electrical and Computer Engineering
University of Southern California, Los Angeles, USA
{shunlinlu,yukezhan,jtliu,pabeerel}@usc.edu

## ABSTRACT

The large number of ReLU non-linearity operations in existing deep networks makes them ill-suited for latency-efficient private inference (PI). Existing techniques to reduce ReLU operations often involve manual effort and sacrifice significant accuracy. In this paper, we first present a novel measure of non-linearity layers' ReLU sensitivity, enabling mitigation of the time-consuming manual efforts in identifying the same. Based on this sensitivity, we then present SENet, a three-stage training method that for a given ReLU budget, automatically assigns per-layer ReLU counts, decides the ReLU locations for each layer's activation map, and trains a model with significantly fewer ReLUs to potentially yield latency and communication efficient PI. Experimental evaluations with multiple models on various datasets show SENet's superior performance both in terms of reduced ReLUs and improved classification accuracy compared to existing alternatives. In particular, SENet can yield models that require up to $\sim 2\times$ fewer ReLUs while yielding similar accuracy. For a similar ReLU budget SENet can yield models with $\sim 2.32\%$ improved classification accuracy, evaluated on CIFAR-100.

## 1 INTRODUCTION

With the recent proliferation of several AI-driven client-server applications including image analysis (Litjens et al., 2017), object detection, speech recognition (Hinton et al., 2012), and voice assistance services, the demand for machine learning inference as a service (MLaaS) has grown. Simultaneously, the emergence of privacy concerns from both the users and model developers has made *private inference* (PI) an important aspect of MLaaS. In PI the service provider retains the proprietary models in the cloud where the inference is performed on the client's encrypted data (*ciphertexts*), thus preserving both model (Kundu et al., 2021b) and data-privacy (Yin et al., 2020).

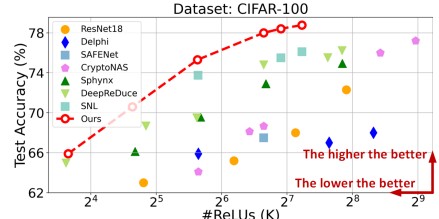

Existing PI methods rely on various cryptographic protocols, including homomorphic encryption (HE) (Brakerski & Vaikuntanathan, 2014; Gentry, 2009) and additive secret sharing (ASS) (Goldreich et al., 2019) for the linear operations in the convolutional and fully connected (FC) layers. For example, popular methods like Gazelle (Juvekar et al., 2018), DELPHI (Mishra et al., 2020), and Cheetah (Reagen et al., 2021) use HE while MiniONN (Liu et al., 2017) and CryptoNAS (Ghodsi et al., 2020) use ASS. For performing the non-linear ReLU operations, the PI methods generally use Yao's Garbled Circuits (GC) (Yao, 1986). However, GCs demand orders of magnitude higher latency and communication than the PI of linear operations, making latency-efficient PI an exceedingly difficult task. In contrast, standard inference latency is dominated by the linear operations (Kundu et al., 2022b) and is significantly lower than that of PI.

This has motivated the unique problem of reducing the number of ReLU non-linearity operations to reduce the communication and latency overhead of PI. In particular, recent literature has leveraged

Figure 1: Comparison of various methods in accuracy vs. #ReLU trade-off plot. SENet outperforms the existing approaches with an accuracy improvement of up to $\sim 4.5\%$ for similar ReLU budget.

---

[*]Part of the work was done when the first author was a graduate student at USC.

Table 1: Comparison between existing approaches in yielding efficient models to perform PI. Note, SENet++ can yield a model that can be switched to sub-models of reduced channel sizes.

| Name | Method used | Reduced non-linearity | Granularity | Reduce model dimension | Supports dynamic channel dropping |
|---|---|---|---|---|---|
| Irregular pruning | Various | ✗ | Scalar weight | ✗ | ✗ |
| Structured pruning | Various | ✓ | Channel, filter | ✓ | ✗ |
| Sphynx (Cho et al., 2021) | NAS | ✓ | Layer-block | ✗ | ✗ |
| CryptoNAS (Ghodsi et al., 2020) | NAS | ✓ | Layer-block | ✗ | ✗ |
| DELPHI (Mishra et al., 2020) | NAS + PA | ✓ | Layer-block | ✗ | ✗ |
| SAFENet (Lou et al., 2021) | NAS + PA | ✓ | Channel | ✗ | ✗ |
| DeepReDuce (Jha et al., 2021) | Manual + HE | ✓ | Layer-block | ✗ | ✗ |
| SNL (Cho et al., 2022) | $l_1$-regularized | ✓ | Channel, pixel | ✗ | ✗ |
| SENet (ours) | Automated | ✓ | Channel, pixel | ✗ | ✗ |
| SENet++ (ours) | Automated | ✓ | Channel, pixel | ✓ | ✓ |

neural architecture search (NAS) to optimize both the number and placement of ReLUs (Ghodsi et al., 2020; Cho et al., 2021). However, these methods often cost significant accuracy drop, particularly when the ReLU budget is low. For example, with a ReLU budget of 86k, CryptoNAS costs ∼9% accuracy compared to the model with all ReLUs (AR) present. DeepReDuce (Jha et al., 2021) used a careful multi-stage optimization and provided reduced accuracy drop of ∼3% at similar ReLU budgets. However, DeepReDuce heavily relies on manual effort for the precise removal of ReLU layers, making this strategy exceedingly difficult, particularly, for models with many layers. A portion of these accuracy drops can be attributed to the fact that these approaches are constrained to remove ReLUs at a higher granularity of layers and channels rather than at the pixel level. Only very recently, (Cho et al., 2022) proposed $l_1$-regularized pixel level ReLU reduction. However, such approaches are extremely hyperparameter sensitive and often do not guarantee meeting a specific ReLU budget. Moreover, the large number of training iterations required for improved accuracy may not be suitable for compute-limited servers (Mishra et al., 2020).

**Our contributions.** Our contribution is three-fold. We first empirically demonstrate the relation between a layer's sensitivity towards pruning and its associated ReLU sensitivity. Based on our observations, we introduce an automated layer-wise ReLU sensitivity evaluation strategy and propose SENet, a three-stage training process to yield secure and efficient networks for PI that guarantees meeting a target ReLU budget without any hyperparameter-dependent iterative training. In particular, for a given global ReLU budget, we first determine a sensitivity-driven layer-wise non-linearity (ReLU) unit budget. Given this budget, we then present a layer-wise ReLU allocation mask search. For each layer, we evaluate a binary mask tensor with the size of the corresponding activation map for which a 1 or 0 signifies the presence or absence of a ReLU unit, respectively. Finally, we use the trained mask to create a partial ReLU (PR) model with ReLU present only at fixed parts of the non-linearity layers, and fine-tune it via distillation from an iso-architecture trained AR model. Importantly, we support ReLU mask allocation both at the granularity of individual pixels and activation channels.

To further reduce both linear and non-linear (ReLU) layer compute costs, we extend our approach to SENet++. SENet++ uses a single training loop to train a model of different channel dropout rates (DRs) $d_r$ ($d_r \leq 1.0$) of the weight tensor, where each $d_r$ yields a sub-model with a MAC-ReLU budget smaller than or same as that of the original one. In particular, inspired by the idea of ordered dropout (Horvath et al., 2021), we train a PR model with multiple dropout rates (Horvath et al., 2021), where each dropout rate corresponds to a scaled channel sub-model having number of channels per layer $\propto$ the $d_r$. This essentially allows the server to yield multiple sub-models for different compute requirements that too via a single training loop, without costly memory footprint. Table 1 compares the important characteristics of our methods with existing alternatives.

We conduct extensive experiments and ablations on various models including variants of ResNet, Wide Residual Networks, and VGG on CIFAR-10, CIFAR-100, Tiny-ImageNet, and ImageNet datasets. Experimental results show that SENet can yield SOTA accuracy-ReLU trade-off with an improved accuracy of up to ∼2.32% for similar ReLU budgets. SENet++ ($d_r = 0.5$) can further improve the MAC and ReLU cost of SENet, with an additional saving of $4\times$ and ∼$2\times$, respectively.

## 2 PRELIMINARIES AND RELATED WORK

**Cryptographic primitives.** We briefly describe the relevant cryptographic primitives in this section.

*Additive secret sharing.* Given an element $x$, an ASS of $x$ is the pair $(\langle x \rangle_1, \langle x \rangle_2) = (x - r, r)$, where $r$ is a random element and $x = \langle x \rangle_1 + \langle x \rangle_2$. Since $r$ is random, the value $x$ cannot be revealed by a single share, so that the value $x$ is hidden.

*Homomorphic encryption.* HE (Gentry, 2009) is a public key encryption scheme that supports homomorphic operations on the ciphertexts. Here, encryption function $E$ generates the ciphertext $t$ of a plaintext $m$ where $t = E(m, pk)$, and a decryption function $D$ obtains the plaintext $m$ via $m = D(t, sk)$, where $pk$ and $sk$ are corresponding public and secret key, respectively. In PI, the results of linear operations can be obtained homomorphically through $m_1 \circ m_2 = D(t_1 \star t_2, sk)$, where $\circ$ represents a linear operation, $\star$ is its corresponding homomorphic operation, $t_1$ and $t_2$ are the ciphertexts of $m_1$ and $m_2$, respectively.

*Garbled circuits.* GC (Yao, 1986) allows two parties to jointly compute a Boolean function $f$ over their private inputs without revealing their inputs to each other. The Boolean function $f$ is represented as a Boolean circuit $C$. Here, a garbler creates an encoded Boolean circuit $\tilde{C}$ and a set of input-correspondent labels through a procedure $Garble(C)$ to send $\tilde{C}$ and the labels to the other party who acts as an evaluator. The evaluator further sends the output label upon evaluation via $Eval(\tilde{C})$. Finally, the garbler decrypts the labels to get the plain results to share with the evaluator.

**Private inference.** Similar to (Mishra et al., 2020), in this paper, we focus on a semi-honest client-server PI scenario where a client, holding private data, intends to use inference service from a server having a private model. Specifically, the semi-honest parties strictly follow the protocol but try to reveal their collaborator's private data by inspecting the information they received. On the other hand, a malicious client could deviate from the protocol.

To defend against various threats existing cryptographic protocols (Mishra et al., 2020; Ghodsi et al., 2020; Lou et al., 2021) rely on the popular online-offline topology (Mishra et al., 2020), where the client data independent component is pre-computed in the offline phase (Juvekar et al., 2018; Mishra et al., 2020; Ghodsi et al., 2021; Lehmkuhl et al., 2021). For the linear operations, DELPHI (Mishra et al., 2020) and MiniONN (Liu et al., 2017) move the heavy primitives in HE and ASS to offline enabling fast linear operations during PI online stage. However, the compute-heavy $Eval(\tilde{C})$ of GC keeps the ReLU cost high even at the online stage.

**ReLU reduction for efficient PI.** Existing works use model designing with reduced ReLU counts via either search for efficient models (Mishra et al., 2020; Ghodsi et al., 2020; Lou et al., 2021; Cho et al., 2021) or manual re-design from an existing model (Jha et al., 2021). In particular, SAFENet (Lou et al., 2021) enables more fine-grained channel-wise substitution and mixed-precision activation approximation. CryptoNAS (Ghodsi et al., 2020) re-designs the neural architectures through evolutionary NAS techniques to minimize ReLU operations. Sphynx (Cho et al., 2021) further improves the search by leveraging differentiable macro-search NAS (Liu et al., 2018a) in yielding efficient PI models. DeepReDuce (Jha et al., 2021), on the other hand, reduced ReLU models via a manual effort of finding and dropping redundant ReLU layers starting from an existing standard model. Finally, a recent work (Cho et al., 2022) leveraged $l_1$-regularization to remove ReLU at the pixel level to yield SOTA accuracy vs non-linearity trade-off. However, the extreme hyperparameter dependence of such methods often provide sub-optimal solution and does not necessarily guarantee meeting a target ReLU budget. Moreover, a resource-limited server (Mishra et al., 2020) may not afford costly iterative training (Cho et al., 2022) in reducing the ReLU count.

## 3 MOTIVATIONAL STUDY: RELATION BETWEEN RELU IMPORTANCE AND PRUNING SENSITIVITY

Existing work to find the importance of a ReLU layer (Jha et al., 2021), requires manual effort and is extremely time consuming. In contrast, model pruning literature (Lee et al., 2018; Kundu et al., 2021a) leveraged various metrics to efficiently identify a layer's sensitivity towards a target pruning ratio. In particular, a layer's *pruning sensitivity* can be quantitatively defined as the accuracy reduction caused by pruning a certain ratio of parameters from it (Ding et al., 2019). In particular, recent literature leveraged sparse learning (Ding et al., 2019; Kundu et al., 2021a) and used a trained sparse model to evaluate the sensitivity of a layer $l$ ($\eta_{\boldsymbol{\theta}^l}$) as the ratio $\frac{\text{total \# of non-zero layer parameters}}{\text{total \# layer parameters}}$. **Despite significant progress in weight pruning sensitivity, due to the absence of any trainable parameter in the ReLU layer, its sensitivity for a given ReLU budget is yet to be explored**. We hypothesize that there may be a correlation between a layer's pruning sensitivity (Kundu et al., 2021a) and the importance of ReLU and have conducted the following experiments to explore this.

Let us assume an $L$-layer DNN model $\Phi$ parameterized by $\boldsymbol{\Theta} \in \mathbb{R}^m$ that learns a function $f_\Phi$, where $m$ represents the total number of model parameters.

The goal of DNN parameter pruning is to identify and remove the unimportant parameters from a DNN and yield a reduced parameter model that has comparable performance to the baseline unpruned model. As part of the pruning process for a given parameter density $d$, each parameter is associated with an auxiliary indicator variable $c$ belonging to a mask tensor $\boldsymbol{c} \in \{0, 1\}^m$ such that only those $\theta$ remain non-zero whose corresponding $c = 1$. With these notations, we can formulate the training optimization as

$$\min \mathcal{L}(f_\Phi(\boldsymbol{\Theta} \odot \boldsymbol{c})), \text{ s.t. } ||\boldsymbol{c}||_0 \le d \times m \tag{1}$$

where $\mathcal{L}(.)$ represents the loss function, which for image classification tasks is generally the cross-entropy (CE) loss. We used a sparse learning framework (Kundu et al., 2021a) to train a ResNet18 on CIFAR-100 for a target $d = 0.1$ and computed the pruning sensitivity of each layer. In particular, as shown in Fig. 2, earlier layers have higher pruning sensitivity than later ones. This means that to achieve close to baseline performance, the model trains later layers' parameters towards zero more than those of earlier layers.

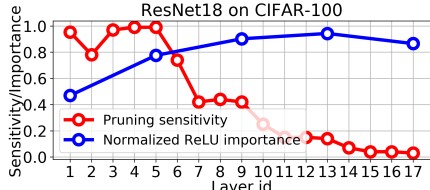

Figure 2: Layer-wise pruning sensitivity ($d = 0.1$) vs. normalized ReLU importance. The later layers are less sensitive to pruning, and, thus, can afford significantly more zero-valued weights as opposed to the earlier ones. On the contrary, later ReLU stages generally have higher importance.

We then compared this trend with that of the importance of different ReLU layers as defined in (Jha et al., 2021). In particular, we first identified five different modules of ReLU placement in a ResNet18, the pre-basic-block (BB) stem, BB1, BB2, BB3, and BB4. We then created five ResNet18 variants with ReLU nonlinearity present only at one of the modules while replacing non-linearity at the other modules with identity layers. We identify the modules yielding higher accuracy to be the ones with higher ReLU importance (Jha et al., 2021). We then normalized the importance of a ReLU stage with accuracy Acc as the ratio $(\text{Acc} - \text{Acc}_{min})/(\text{Acc}_{max} - \text{Acc}_{min})$. Here $\text{Acc}_{max}$ and $\text{Acc}_{min}$ correspond to the accuracy of models with all and no ReLUs, respectively.

As depicted in Figure 2, the results show that the ReLU importance and parameter pruning sensitivity of a layer are inversely correlated. This inverse correlation may imply that a pruned layer can afford to have more zero-valued weights when the associated ReLU layer forces most of the computed activation values to zero.

## 4  SENet Training Methodology

As highlighted earlier, for a large number of ReLU layers $L_r$, the manual evaluation and analysis of the candidate architectures become inefficient and time consuming. Moreover, the manual assignment of ReLU at the pixel level becomes even more intractable because the number of pixels that must be considered, explodes. To that end, we now present SENet, a three-stage automated ReLU trimming strategy that can yield models for a given reduced ReLU budget.

### 4.1  Sensitivity Analysis

Inspired by our observations in Section 3, we define the ReLU sensitivity of a layer $l$ as

$$\eta_{\boldsymbol{\alpha}^l} = (1 - \eta_{\boldsymbol{\theta}^l}) \tag{2}$$

It is important to emphasize that, unlike ReLU importance, ReLU sensitivity does not require training many candidate models. However, $\eta_{\boldsymbol{\theta}^l}$ can only be evaluated for a specific $d$. We empirically observe that $d > 0.3$ tends to yield uniform sensitivity across layers due to a large parameter budget. In contrast, ultra-low density $d < 0.1$, costs non-negligible accuracy drops (Liu et al., 2018b; Kundu et al., 2022a). Based on these observations, we propose to quantify ReLU sensitivity with a *proxy density* of $d = 0.1$.

Moreover, to avoid the compute-heavy pruning process, we leverage the idea of sensitivity evaluation before training (Lee et al., 2018). On a sampled mini batch from training data $\mathcal{D}$, the sensitivity of the $j^{th}$ connection with associated indication variable and vector as $c_j$ and $\boldsymbol{e}_j$, can be evaluated as,

---

**Algorithm 1:** Layer-wise #ReLU Allocation Algorithm

---

**Data:** Global ReLU budget $r$, model parameters $\boldsymbol{\Theta}$, model parameter proxy density $d$,

1  number of ReLU layers $L_r$, active ReLU indicator $\boldsymbol{a} \in \{1\}^{L_r}$
2  **Output:** Per-layer # ReLU count.
3  $\boldsymbol{\eta_\alpha} \leftarrow$ `evalActSens`$(\boldsymbol{\Theta}, d)$
4  **for** $l \leftarrow 0$ **to** $L_r$ **do**
5  $\quad \eta_{\boldsymbol{\alpha}^l} \leftarrow \dfrac{\eta_{\boldsymbol{\alpha}^l}}{\sum_{i=0}^{L} \eta_{\boldsymbol{\alpha}^i} \times a^i}$
6  **end**
7  `initVals`$(r_{remain}, r_{total}, \boldsymbol{r}_{final})$
8  **while** $r_{total} < r$ **do**
9  $\quad$ **for** $l \leftarrow 0$ **to** $L$ **do**
10  $\quad\quad r_{cur}^l \leftarrow$ `assignReluProportion`$(r_{remain}, \eta_{\boldsymbol{\alpha}^l}, \boldsymbol{a})$
11  $\quad\quad r_{final}^l, r_{total} \leftarrow$ `assignUpdateRelu`$(r_{final}^l, r_{cur}^l, r_{total})$
12  $\quad$ **end**
13  **end**
14  $r_{remove} \leftarrow r_{total} - r$
15  **while** $r_{remove} > 0$ **do**
16  $\quad$ **for** $l \leftarrow 0$ **to** $L$ **do**
17  $\quad\quad r_{cur}^l \leftarrow$ `removeReluProportion`$(r_{del}, \eta_{\boldsymbol{\alpha}^l}, \boldsymbol{a})$
18  $\quad\quad r_{final}^l, r_{remove} \leftarrow$ `removeUpdateRelu`$(r_{final}^l, r_{cur}^l, r_{remove})$
19  $\quad$ **end**
20  **end**
21  `return` $\boldsymbol{r}_{final}$

---

$$\Delta\mathcal{L}_j(f_\Phi(\boldsymbol{\Theta};\mathcal{D})) = g_j(f_\Phi(\boldsymbol{\Theta};\mathcal{D})) = \frac{\partial \mathcal{L}(f_\Phi(\boldsymbol{c}\odot\boldsymbol{\Theta};\mathcal{D}))}{\partial c_j}\bigg|_{\boldsymbol{c}=\boldsymbol{1}} \tag{3}$$
$$= \lim_{\delta\to 0} \frac{\mathcal{L}(f_\Phi(\boldsymbol{c}\odot\boldsymbol{\Theta};\mathcal{D})) - \mathcal{L}(f_\Phi((\boldsymbol{c}-\delta\boldsymbol{e}_j)\odot\boldsymbol{\Theta};\mathcal{D}))}{\delta}\bigg|_{\boldsymbol{c}=\boldsymbol{1}}$$

where $\boldsymbol{c}$ is a vector containing all indicator variables. The $\frac{\partial \mathcal{L}}{\partial c_j}$ is an infinitesimal version of $\Delta\mathcal{L}_j$ measuring the impact of a change in $c_j$ from $1 \to 1 - \delta$. It can be computed using one forward pass for all $j$ at once. We normalize the connection sensitivities, rank them, and identify the top d-fraction of connections. We then define the layer sensitivity $\eta_{\boldsymbol{\Theta}^l}$ as the fraction of connections of each layer that are in the top d-fraction. For a given global ReLU budget $r$, we then assign the # ReLU for each layer proportional to its normalized ReLU sensitivity. The details are shown in Algorithm 1 (Fig. 3 as point ①). Note $r_{final}^l$ in Algorithm 1 represents the allocated #ReLUs of layer $l$ at the end of stage 1, with $\boldsymbol{r}_{final}$ representing the set of #ReLUs for all the ReLU layers.

## 4.2 ReLU Mask Identification

After layer-wise #ReLU allocation, we identify the ReLU locations in each layer's activation map. In particular, for a non-linear layer $l$, we assign a mask tensor $M^l \in \{0,1\}^{h^l \times w^l \times c^l}$, where $h^l, w^l$, and $c^l$ represents the height, width, and the number of channels in the activation map. For a layer $l$, we initialize $M$ with $r_{final}^l$ assigned 1's with random locations. Then we perform a distillation-based training of the PR model performing ReLU ops only at the locations of the masks with 1, while distilling knowledge from an AR model of the same architecture (see Fig. 3, point ②). At the end of each epoch, for each layer $l$, we rank the top-$r_{final}^l$ locations based on the highest absolute difference between the PR and AR model's post-ReLU activation output (averaged over all the mini-batches) for that layer, and update the $M^l$ with 1's at these locations. This, on average, de-emphasizes the locations where the post-ReLU activations in both the PR and AR models are positive. We terminate mask evaluation once the ReLU mask[1] evaluation reaches the maximum mask training epochs or when the normalized hamming distance between masks generated after two consecutive epochs is below a certain pre-defined $\epsilon$ value. **Notably, there has been significant research in identifying important trainable parameters (Savarese et al., 2020; Kusupati et al., 2020; Kundu et al., 2020; 2022c;b; Babakniya et al., 2022) through various proxies including magnitude, gradient, Hessian, however, due to the absence of any trainable parameter in the ReLU layer, such methods can't be deployed in identifying important ReLU units of a layer.**

---

[1]The identified mask tensor has non-zeros irregularly placed. This can be easily extended to the generation of the structured mask, by allowing the assignment and removal of mask values at the granularity of channels instead of activation scalar (Kundu et al., 2021a).

Figure 3: Different stages of the proposed training methodology for efficient private inference with dynamic channel reduction. For example, the model here supports two channel SFs, $S_1$ and $S_2$. Note, similar to (Horvath et al., 2021), for each SF support we use a separate batch-normalization (BN) layer to maintain separate statistics.

**Channel-wise ReLU mask identification.** The mask identification technique described above, creates irregular ReLU masks. To support a coarser level of granularity where the ReLU removal happens "channel-wise", we now present a simple yet effective extension of the mask identification. For a layer $l$, we first translate the total non-zero ReLU counts to total non-zero ReLU channels as $r_c^l = \lceil \frac{r_{final}^l}{h^l w^l} \rceil$. We then follow the same procedure as irregular mask identification, however, only keep top-$r_c^l$ channels as non-zero.

## 4.3 MAXIMIZING ACTIVATION SIMILARITY VIA DISTILLATION

Once the mask for each layer is frozen, we start our final training phase in which we maximize the similarity between activation functions of our PR and AR models, see Fig. 3, point ③. In particular, we initialize a PR model with the weights and mask of best PR model of stage 2 and allow only the parameters to train. We train the PR model with distillation via KL-divergence loss (Hinton et al., 2015; Kundu & Sundaresan, 2021) from a pre-trained AR along with a CE-loss. Moreover, we introduce an AR-PR post-ReLU activation mismatch (PRAM) penalty into the loss function. This loss drives the PR model to have activation maps that are similar to that of the AR model.

More formally, let $\Psi_{pr}^m$ and $\Psi_{ar}^m$ represent the $m^{th}$ pair of vectorized post-ReLU activation maps of same layer for $\Phi_{pr}$ and $\Phi_{ar}$, respectively. Our loss function for the fine-tuning phase is given as

$$\mathcal{L} = (1-\lambda) \underbrace{\mathcal{L}_{pr}(y, y^{pr})}_{\text{CE loss}} + \lambda \underbrace{\mathcal{L}_{KL}\left(\sigma\left(\frac{z^{ar}}{\rho}\right), \sigma\left(\frac{z^{pr}}{\rho}\right)\right)}_{\text{KL-div. loss}} + \frac{\beta}{2} \sum_{m \in I} \underbrace{\left\| \frac{\Psi_{pr}^m}{\|\Psi_{pr}^m\|_2} - \frac{\Psi_{ar}^m}{\|\Psi_{ar}^m\|_2} \right\|_2}_{\text{PRAM loss}} \quad (4)$$

where $\sigma$ represents the softmax function with $\rho$ being its temperature. $\lambda$ balances the importance between the CE and KL divergence loss components, and $\beta$ is the weight for the PRAM loss. Similar to (Zagoruyko & Komodakis, 2016a), we use the $l_2$-norm of the normalized activation maps to compute this loss.

## 4.4 SENET++: SUPPORT FOR ORDERED CHANNEL DROPPING

To yield further compute-communication benefits, we now present an extension of SENet, namely SENet++, that can perform the ReLU reduction while also supporting inference with reduced model sizes. In particular, we leverage the idea of ordered dropout (OD) (Horvath et al., 2021) to simultaneously train multiple sub-models with different fractions of channels. The OD method is parameterized by a candidate dropout set $\mathcal{D}_r$ with dropout rate values $d_r \in (0, 1]$. At a selected $d_r$ for any layer $l$, the model uses a $d_r$-sub-model with only the channels with indices $\{0, 1, ..., \lceil d_r \cdot C_l \rceil - 1\}$ active, effectively pruning the remaining $\{\lceil d_r \cdot C_l \rceil, ..., C_l - 1\}$ channels. Hence, during training, the selection of a $d_r$-sub-model with $d_r < 1.0 \in \mathcal{D}_r$, is a form of channel pruning, while $d_r = 1.0$ trains the full model. For each mini-batch of data, we perform a forward pass once for each value of $d_r$ in $\mathcal{D}_r$, accumulating the loss. We then perform a backward pass in which the model parameters are updated based on the gradients computed on the accumulated loss. We first train an AR model with a dropout set $\mathcal{D}_r$. For the ReLU budget evaluation, we consider only the model with $d_r = 1.0$, and finalize the mask by following the methods in Sections 4.1 and 4.2. During the maximizing of activation similarity stage, we fine-tune the PR model supporting the same set $\mathcal{D}_r$ as that of the AR model. In particular, the loss function for the fine-tuning is the same as 4, for $d_r = 1.0$. For $d_r < 1.0$, we exclude the PRAM loss because we empirically observed that adding the PRAM loss

for each sub-model on average does not improve accuracy. During inference, SENet++ models can be dynamically switched to support reduced channel widths, reducing the number of both ReLUs and MACs compared to the baseline model.

## 5 EXPERIMENTS

### 5.1 EXPERIMENTAL SETUP

**Models and Datasets.** To evaluate the efficacy of the SENet yielded models, we performed extensive experiments on three popular datasets, CIFAR-10, CIFAR-100 (Krizhevsky et al., 2009), Tiny-ImageNet (Hansen, 2015), and ImageNet[2] with three different model variants, namely ResNet (ResNet18, ResNet34) (He et al., 2016), wide residual network (WRN22-8) (Zagoruyko & Komodakis, 2016b), and VGG (VGG16) (Simonyan & Zisserman, 2014). We used PyTorch API to define and train our models on an Nvidia RTX 2080 Ti GPU.

**Training Hyperparameters.** We performed standard data augmentation (horizontal flip and random cropping with reflective padding) and the SGD optimizer for all training. We trained the baseline all-ReLU model for 240, 120, and 60 epochs for CIFAR, Tiny-ImageNet, and ImageNet respectively, with a starting learning rate (LR) of 0.05 that decays by a factor of 0.1 at the 62.5%, 75%, and 87.5% training epochs completion points. For all the training we used an weight decay coefficient of $5 \times 10^{-4}$. For a target ReLU budget, we performed the mask evaluation for

Table 2: Runtime and communication costs of linear and ReLU operations for 15-bit fixed-point model parameters/inputs and 31-bit ReLUs (Mishra et al., 2020).

| Operation | Mode | Runtime($\mu s$) | Comm. cost(KB) |
|---|---|---|---|
| Linear | Offline | 32.6 | 0.095 |
| | Online | 0.248 | 0.000563 |
| ReLU | Offline | 154.9 | 17.5 |
| | Online | 85.3 | 2.048 |

150, 100, and 30 epochs, respectively, for the three dataset types with the $\epsilon$ set to 0.05, meaning the training prematurely terminates when less than 5% of the total #ReLU masks change their positions. Finally, we performed the post-ReLU activation similarity improvement for 180, 120, and 50 epochs, for CIFAR, Tiny-ImageNet, and ImageNet respectively. Also, unless stated otherwise, we use $\lambda = 0.9$, and $\beta = 1000$ for the loss described in Eq. 4. Further details of our training hyper-parameter choices are provided in the Appendix. In Table 5, we report the accuracy averaged over three runs.

### 5.2 SENET RESULTS

As shown in Table 5, SENet yields models that have higher accuracy than existing alternatives by a significant margin while often requiring fewer ReLUs. For example, at a small ReLU budget of $\leq$ 100k, our models yield up to 4.15% and 7.8% higher accuracy, on CIFAR-10 and CIFAR-100, respectively. At a ReLU budget of $\leq$ 500k, our improvement is up to 0.50% and 2.38%, respectively, on the two datasets. We further evaluate the communication saving due to the nonlinearity reduction by taking the per ReLU communication cost mentioned in Table 2. In particular, the communication saving reported in the $8^{th}$ column of Table 5 is computed as the ratio of communication costs associated with an AR model to that of the corresponding PR model with reduced Re-

Table 3: Results on Tiny-ImageNet and ImageNet.

| Model | Baseline Acc% | #ReLU (k) | Method | Test Acc% | Acc%/ #1k ReLU | Comm. Savings |
|---|---|---|---|---|---|---|
| | | | Dataset: Tiny-ImageNet | | | |
| ResNet18 | 66.1 | 142 | SENet | 58.9 | 0.414 | ×15.7 |
| | | 298 | | **64.96** | 0.218 | 7.5× |
| | | 393 | DeepReDuce(Jha et al., 2021) | 61.65 | 0.157 | 5.7× |
| | | 917 | | 64.66 | 0.071 | ×2.4 |
| | | | Dataset: ImageNet | | | |
| ResNet18 | 71.94 | 600 | SENet | 70.28 | 0.117 | 3.86× |
| | | 950 | | **71.16** | 0.075 | 2.43× |

Table 4: Results with ReLU reduction at the granularity of activation channel evaluated on CIFAR-100.

| Model | Baseline Acc% | #ReLU (k) | Method | Test Acc% | Acc%/ #1k ReLU | Comm. Savings |
|---|---|---|---|---|---|---|
| WRN22-8 | 80.82 | **180** | SENet | 79.02 | 0.44 | **7.7×** |
| | | 240 | SENet | **79.3** | 0.33 | 5.8× |
| | | 200 | SNL (Cho et al., 2022) | 77.45 | 0.38 | 6.9× |

LUs. We did not report any saving for the custom models, as they do not have any corresponding AR baseline model. On Tiny-ImageNet, SENet models can provide up to 0.3% higher performance while requiring 3.08× fewer ReLUs (Table 3). More importantly, even for a high resolution dataset like ImageNet, SENet models can yield close to the baseline performance, depicting the efficacy of our proposed training method.

---

[2]On ImageNet, for comprehensive training with limited resources, we sample 100 classes from the ImageNet dataset with 500 and 50 training and test examples per class, respectively.

Table 5: Performance of SENet and other methods on various datasets and models.

| min ≤ r ≤ max | Model | Baseline Acc% | #ReLU (k) | Method | Test Acc% | Acc%/ #1k ReLU | Comm. Savings |
|---|---|---|---|---|---|---|---|
| | | | | Dataset: CIFAR-10 | | | |
| | VGG16 | 93.8 | 12.5 | | 91.6 | 7.33 | 23.6× |
| | | | 49.2 | SENet(ours) | 93.16 | 1.89 | 6.0× |
| 0 ≤ r ≤ 100k | ResNet18 | 95.2 | 49.1 | | 93.60 | 1.9 | 11.3× |
| | | | 82 | | **93.05** | 1.14 | 6.8× |
| | ResNet18 | 95.2 | 12.9 | SNL (Cho et al., 2022) | 88.23 | 6.84 | 43.1× |
| | VGG16 | 93.8 | 36.8 | DeepReDuce (Jha et al., 2021) | 88.9 | 2.41 | 8× |
| | VGG16 | 93.8 | 126 | SENet(ours) | 93.42 | 0.74 | 2.3× |
| | ResNet18 | 95.2 | 150 | | **94.91** | 0.63 | 3.7× |
| 100k ≤ r ≤ 500k | VGG16 | 93.8 | 126 | DeepReDuce (Jha et al., 2021) | 92.5 | 0.73 | 2.3× |
| | VGG16 | 93.8 | 126 | SAFENet (Lou et al., 2021) | 88.9 | 0.7 | 2.3× |
| | Custom Net | 95.0 | 100 | CryptoNAS (Ghodsi et al., 2020) | 92.18 | 0.92 | – |
| | | | 500 | | 94.41 | 0.19 | – |
| | | | | Dataset: CIFAR-100 | | | |
| | ResNet18 | 78.05 | 24.6 | | 70.59 | 2.87 | 21.8× |
| | | | 49.6 | | 75.28 | 1.52 | 11.2× |
| | | | 100 | SENet(ours) | **77.92** | 0.78 | 5.6× |
| 0 ≤ r ≤ 100k | ResNet34 | 78.42 | 50.1 | | 74.84 | 1.5 | 19.3× |
| | | | 80 | | 76.66 | 0.96 | 12.1× |
| | ResNet18 | 78.05 | 28.7 | DeepReDuce (Jha et al., 2021) | 68.6 | 2.39 | 19.4× |
| | | | 49.2 | | 69.5 | 1.41 | 11.3× |
| | Custom Net | 74.93 | 51 | Sphynx (Cho et al., 2021) | 69.57 | 1.36 | – |
| | ResNet18 | 78.05 | 150 | | 78.32 | 0.52 | 3.7× |
| | ResNet34 | 78.425 | 200 | | 78.8 | 0.4 | 4.8× |
| | WRN22-8 | 80.82 | 180 | SENet(ours) | 79.12 | 0.44 | 7.7× |
| | | | 240 | | 79.81 | 0.33 | 5.8× |
| | | | 300 | | **80.54** | 0.27 | 4.6× |
| 100k ≤ r ≤ 500k | WRN22-8 | 80.82 | 180 | SNL (Cho et al., 2022) | 77.65 | 0.43 | 7.7× |
| | ResNet18 | 78.05 | 229.4 | DeepReDuce (Jha et al., 2021) | 76.22 | 0.33 | 2.4× |
| | Custom Net | 74.93 | 102 | Sphynx (Cho et al., 2021) | 72.9 | 0.714 | – |
| | | | 230 | | 74.93 | 0.32 | – |
| | Custom Net | 79.07 | 100 | CryptoNAS (Ghodsi et al., 2020) | 68.67 | 0.69 | – |
| | | | 500 | | 77.69 | 0.16 | – |

**Results with activation channel level ReLU reduction.** As shown in Table 4, while trimming ReLUs at a higher granularity of activation channel level, SENet models suffer a little more drop in accuracy compared to that at pixel level. For example, at a ReLU budget of 240k, channel-level ReLU removal yields an accuracy of 79.3% compared to 79.81% of pixel-level. However, compared to existing alternatives, SENet can achieve improved performance of up to 1.85% for similar ReLUs.

## 5.3 SENET++ RESULTS

For SENet++, we performed experiments with $\mathcal{D}_r = [0.5, 1.0]$, meaning each training loop can yield models with two different channel dropout rates. The 0.5-sub-model enjoys a $\sim 4\times$ MACs reduction compared to the full model. Moreover, as shown in Fig. 4, the 0.5-sub-model also requires significantly less #ReLUs due to reduced model size. In particular, the smaller models have #ReLUs reduced by a factor of $2.05\times$, $2.08\times$, and $1.88\times$ on CIFAR-10, CIFAR-100, and Tiny-ImageNet, respectively, compared to the PR full models, averaged over four experiments with different ReLU budgets for each dataset. *Lastly, the similar performance of the SENet and SENet++ models at $d_r = 1.0$ with similar ReLU budgets, clearly depicts the ability of SENet++ to yield multiple sub-models without sacrificing any accuracy for the full model.*

## 5.4 ANALYSIS OF LINEAR AND RELU INFERENCE LATENCY

Table 2 shows the GC-based online ReLU operation latency is $\sim 343\times$ higher than one linear operation (multiply and accumulate), making the ReLU operation latency the dominant latency component. Inspired by this observation, we quantify the online PI latency as that of the $N$ ReLU operations for a model with ReLU budget of $N$. In particular, based on this evaluation, Fig. 5(a) shows the superiority of SENet++ of up to $\sim 9.6\times$ ($\sim 1.92\times$) reduced online ReLU latency on CIFAR-10 (CIFAR-100). With negligibly less accuracy this latency improvement can be up to $\sim 21\times$. Furthermore, when $d_r < 1.0$, SENet++ requires fewer MACs and the linear operation latency can be significantly reduced, as demonstrated in Fig. 5(b).

## 5.5 ABLATION STUDIES

**Importance of ReLU sensitivity.** To understand the importance of layer-wise ReLU sensitivity evaluations at a given ReLU budget, we conducted experiments with evenly allocated

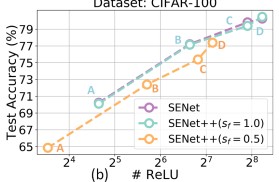 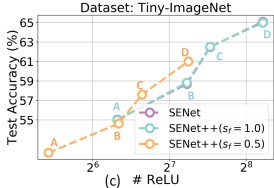

Figure 4: Performance of SENet++ on three datasets for various #ReLU budgets. The points labeled A, B, C, and D corresponds to experiments of different target #ReLUs for the full model ($d_r = 1.0$). For SENet++, note that a single training loop yields two points with the same label corresponding to the two different dropout rates.

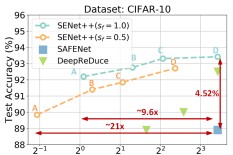 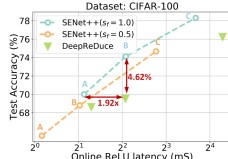 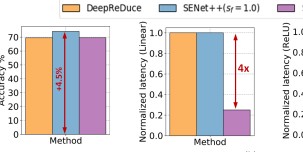

Figure 5: Performance comparison of SENet++ (with $d_r = 1.0$ and 0.5) vs. existing alternatives (a) with VGG16 and ResNet18 in terms of ReLU latency. The labels A, B, C, D correspond to experiments of different target #ReLUs for the full model ($d_r = 1.0$). For SENet++, note that a single training loop yields two points with the same label corresponding to the two different dropout rates. (b) Comparison between DeepReDuce and SENet++ for a target # ReLU budget of ∼50k with ResNet18 on CIFAR-100.

ReLUs. Specifically, for ResNet18, for a ReLU budget of 25% of that of the original model, we randomly removed 75% ReLUs from each PR layer with identity elements to create the ReLU mask, and trained the PR model with this mask. We further trained two other

Table 6: Importance of ReLU sensitivity.

| Model | Baseline Acc% | #ReLU (k) | ReLU Sensitivity | Test Acc% | Acc%/ #1k ReLU | Comm. Savings |
|---|---|---|---|---|---|---|
| ResNet18 | 78.05 | 139.2 | ✗ | 70.12 | 0.503 | ×4 |
| | | 135 | ✓ | **75.88** | 0.56 | ×4.12 |
| | | 70.4 | ✓ | 73.03 | 1.03 | **×7.9** |

PR ResNet18 with similar and lower # ReLU budgets with the per-layer ReLUs assigned following the proposed sensitivity. As shown in Table 6, the sensitivity-driven PR models can yield significantly improved performance of ∼5.76% for similar ReLU budget, demonstrating the importance of proposed ReLU sensitivity.

**Choice of the hyperparameter $\lambda$ and $\beta$.** To determine the influence of the AR teacher's influence on the PR model's learning, we conducted the final stage distillation with $\lambda \in [0.1, 0.3, 0.5, 0.7, 0.9]$ and $\beta \in [100, 300, 500, 700, 1000]$. As shown in Fig. 6, the performance of the student PR model improves with the increasing influence of the teacher both in terms of high $\lambda$ and $\beta$ values. However, we also observe, the performance improvement tends to saturate at $\beta \approx 1000$. Note, we keep $\lambda = 0.5$ and $\beta = 1000$, for the $\beta$ and $\lambda$ ablations, respectively.

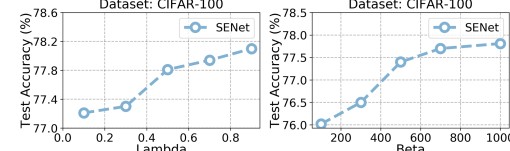

Figure 6: Ablation studies with different $\lambda$ and $\beta$ values for the loss term in Eq. 4.

## 6 CONCLUSIONS

In this paper, we introduced the notion of ReLU sensitivity for non-linear layers of a DNN model. Based on this notion, we present an automated ReLU allocation and training algorithm for models with limited ReLU budgets that targets latency and communication-efficient PI. The resulting networks can achieve similar to SOTA accuracy while significantly reducing the # ReLUs by up to 9.6× on CIFAR-10, enabling a dramatic reduction of the latency and communication costs of PI. Extending this idea of efficient PI to vision transformer models is an interesting future research.

## 7 ACKNOWLEDGMENT

This work was supported in parts by Intel, DARPA under the agreement numbers HR00112190120, and FA8650-18-1-7817.

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

# A  APPENDIX

## A.1  TRAINING HYPERPARAMETERS AND MODELS

The training hyperparameter details for each stage is provided in table 7. Also, for the KL-divergence loss in stage 2 and 3, we fix the temperature value $\rho = 4.0$. To evaluate $\eta_\theta$, we a set of 1000 randomly selected training image samples.

Table 8 details the loss function used for different OD rate values for the SENet++ (at the fine-tuning stage of the PR model i.e. training stage 3). Note, the training stage 1 where the AR model gets trained, we use standard CE loss for all the sub-models with different OD rate values. The detailed fine-tune training algorithm is provided in 2. Note, for SENet, the algorithm remains the same with $\mathcal{D}_r = [1.0]$, meaning supporting only the full model with all the channels present[3].

**Model Selection**. For lower resolution images (CIFAR-10, CIFAR-100, Tiny-ImageNet) compared to that of ImageNet, we have used the variant of ResNet18 and ResNet34 models that are suitable for supporting lower resolution datasets. In particular, we replaced the $7 \times 7$ kernel, stride 2, and padding 3 of the first layer with a $3 \times 3$ kernel having a stride and padding of 1 each. We would also like to highlight that this is a popular practice and can be seen in various other peer-reviewed manuscripts (Wang et al., 2020; Liu et al., 2020; Wong et al., 2020). More importantly, both the existing state-of-the-art methods (Cho et al., 2022; Jha et al., 2021) used similar ResNet models as ours, for their evaluations at a reduced ReLU budget.

Table 7: Hyperparameter settings of SENet/SENet++ training method.

| Model(s) | Dataset | Epoch | | | batch -size | Initial LR | | | Momen- tum | Optim- izer | Weight decay |
|---|---|---|---|---|---|---|---|---|---|---|---|
| | | stage1 | stage2 | stage3 | | stage1 | stage2 | stage3 | | | |
| ResNet18, VGG16 | CIFAR-10 | 240 | 150 | 180 | 128 | 0.05 | 0.05 | 0.01 | 0.9 | SGD | 0.0005 |
| ResNet{18, 34} WRN22-8 | CIFAR-100 | 240 | 150 | 180 | 128 | 0.05 | 0.05 | 0.01 | 0.9 | SGD | 0.0005 |
| ResNet18 | Tiny-ImageNet | 120 | 100 | 120 | 32 | 0.05 | 0.05 | 0.01 | 0.9 | SGD | 0.0005 |
| ResNet18 | ImageNet | 60 | 30 | 50 | 16 | 0.05 | 0.05 | 0.01 | 0.9 | SGD | 0.0005 |

---

[3]We have open-sourced the validation codes with the supplementary.

Table 8: Loss function used for sub-models with different OD rates in SENet++ at training stage 3.

| OD rate ($d_r$) | Loss |
|---|---|
| 1.0 | $(1-\lambda)\mathcal{L}_{CE} + \lambda\,\mathcal{L}_{KL} + \frac{\beta}{2}\,\mathcal{L}_{PRAM}$ |
| < 1.0 | $(1-\lambda)\mathcal{L}_{CE} + \lambda\,\mathcal{L}_{KL}$ |

---

**Algorithm 2:** SENet++ Fine-Tune Training Stage

**Data:** Trained AR model parameters $\boldsymbol{\Theta}_{AR}$, PR model parameters $\boldsymbol{\Theta}_{PR}$, mini-batch size $\mathcal{B}$, learned ReLU mask from stage 2 $\boldsymbol{\Pi}$, OD set $\mathcal{D}_r$.

1 , **Output:** Trained PR model with # ReLU count $r$.
2 $\boldsymbol{\Theta}_{PR} \leftarrow \texttt{applyModelWeight}(\boldsymbol{\Theta}_{AR}, \boldsymbol{\Pi})$
3 **for** $i \leftarrow 0$ **to to** $ep$ **do**
4    **for** $j \leftarrow 0$ **to** $n_{\mathcal{B}}$ **do**
5      **for** *dropout rate $d_r$ in sorted $\mathcal{D}_r$* **do**
6        $\texttt{sampleSBN}(d_r)$ //sample the BN corresponding to $d_r$
7        $\mathcal{L}_{CE} \leftarrow \texttt{computeCELoss}(\boldsymbol{X}_{0:\mathcal{B}}, \boldsymbol{Y}_{0:\mathcal{B}}, \boldsymbol{Y}_{0:\mathcal{B}}^{pr}, \boldsymbol{\Pi})$
8        $\mathcal{L}_{KL} \leftarrow \texttt{computeKLLoss}(f_\Phi(\Theta_{PR}), f_\Phi(\Theta_{AR}), \rho, \boldsymbol{\Pi})$
9        **if** $d_r == 1.0$ **then**
10          $\mathcal{L}_{PRAM} \leftarrow \texttt{computePRAMLoss}(f_\Phi(\Theta_{PR}), f_\Phi(\Theta_{AR}), \boldsymbol{\Pi})$
11          $\mathcal{L} \leftarrow (1-\lambda)\mathcal{L}_{CE} + \lambda\mathcal{L}_{KL} + \frac{\beta}{2}\mathcal{L}_{PRAM}$
12        **else**
13          $\mathcal{L} \leftarrow (1-\lambda)\mathcal{L}_{CE} + \lambda\mathcal{L}_{KL}$
14        **end**
15        $\texttt{accumulateGrad}(\mathcal{L})$
16      **end**
17      $\texttt{updateParam}(\boldsymbol{\Theta}_{PR}, \nabla_{\mathcal{L}})$
18    **end**
19 **end**

---

## A.2 MODEL LATENCY ESTIMATION

Similar to the existing literature (Jha et al., 2021; Cho et al., 2022; Lou et al., 2021), we assume the popular PI framework described in Delphi (Mishra et al., 2020), and leverage their reported per ReLU operation latency to estimate the total online cryptographic private inference latency. In particular, assuming sequential execution, the total latency can be estimated as

$$T = (N_{MAC} * t_{mac} + N_{ReLU} * t_{relu})\mu S. \tag{5}$$

where $N_{MAC}$ and $N_{ReLU}$ corresponds to the total number of MAC and ReLU operations required to perform a single forward pass. $t_{relu}$ represents per ReLU cipher-text execution time and its value is assumed as $85.3\mu S$. Note, Per ReLU cipher text execution wall-clock time is taken from (Mishra et al., 2020) and the execution of 1000 ReLUs takes proportional time as reported in (Cho et al., 2022). Thus similar to (Cho et al., 2022; Jha et al., 2021), we extrapolated per-ReLU wall clock time to extract total ReLU latency for cryptographic inference. Similarly, we evaluated the per linear operation latency $t_{mac}$ to be $0.248\mu S$ (Mishra et al., 2020). Now, as cryptographic ReLU latency is around $343\times$ costlier than that for linear ops, similar to earlier literature (Jha et al., 2021; Cho et al., 2022; Lou et al., 2021), for comparable $N_{MAC}$ and $N_{ReLU}$ we can approximate the total latency with that of the non-linear latency. Note, the values of the ReLU and linear operation latency are taken from (Mishra et al., 2020), however, we understand with recent improvement of operation latency the 'per-operation delay' can reduce (Huang et al., 2022).

## A.3 DISCUSSION ON THE RELATION BETWEEN RELU IMPORTANCE AND PRUNING SENSITIVITY

Fig. 2 of the main manuscript demonstrated the inverse trend between ReLU importance and parameter pruning sensitivity. Here, as an exemplary, we used an extremely low target parameter density $d = 0.1$ to compute the parameter sensitivity. This choice forces an aggressive drop in pruning sensitivity of later layers because they correspond to the majority of the model parameters. In contrast, when we set a less aggressive compression with higher parameter density $d$ of 0.3 or 0.5, we observe a more gradual reduction in weight pruning sensitivity at later layers similar to change observed with ReLU importance. More importantly, as shown in Fig. 7, the inverse trend between

ReLU importance and pruning sensitivity can still be clearly observed irrespective of the choice of $d$, while abruptness of sensitivity change remains a function of target $d$.

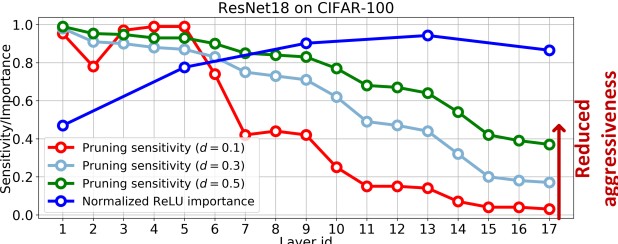

Figure 7: Ablation with different $d$ yielding different layer-wise parameter pruning sensitivity. It can be clearly observed that as the $d$ increases, the aggressiveness of sensitivity change reduces and becomes similar to what we observe for the ReLU importance plot.

## A.4 MORE RESULTS AND ANALYSIS

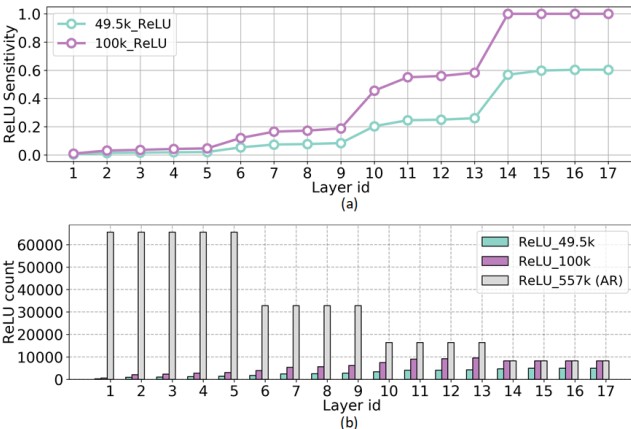

Figure 8: (a) ReLU sensitivity per non-layer layer meeting different target ReLU budget, (b) ReLU count for different layers, evaluated following the sensitivity. We used ResNet18 on CIFAR-100 for this analysis.

**Layer-wise ReLU Sensitivity.** Fig. 8(a) shows the ReLU sensitivity per layer. As elaborated in the Section 3, the ReLU sensitivity is more in the later layers, making our ReLU sensitivity follow similar trend as ReLU importance in DeepReDuce (Jha et al., 2021). Fig. 8(b) shows the layer wise ReLU count for different ReLU budget, allocation driven by sensitivity (Fig. 8(a)). With the increasing count of ReLU budget, the assignment of ReLU happens more aggressively at the later layers, compared to the earlier ones.

**SENet++: Training with More Than 2 Dropout Rates.** In the original manuscript we showed results with two dropout rates ($\mathcal{D}_r = [0.5, 1.0]$), while training with different dropouts, and yield models with different channel width factors. We now show results with $\mathcal{D}_r = [0.25, 0.5, 0.75, 1.0]$. In particular, Fig. 9 and 10 show the results with models yielded via training on *four* different $d_r$ choices. As shown in the Fig. 9(a) and 10(a), the performance of the models at $d_r = 1.0$ and $d_r = 0.5$, are similar for both two and four $d_r$ choices, making SENet++ an efficient algorithm in yielding multiple reduced FLOPs/ReLU models. Moreover, the effective ReLU reduction remains proportional to the corresponding ordered dropout rate (OD rate) (Fig. 9(b) and 10(b)). Fig.9(c) shows the effective FLOPs reduction for the CONV layers while performing inference at a reduced OD rate model selection.

## A.5 FURTHER ABLATION STUDIES

**Ablation Studies for Stage 2 Importance.** To understand the importance of the ReLU mask identification stage, we now present results for SENet with and without that stage. In particular, for the model without stage 2, we randomly assign the ReLU mask by following the ReLU layer sensitivity, meaning layers having higher sensitivity will have non-zero mask values of similar proportions at

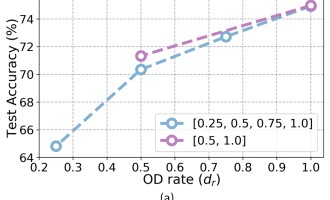 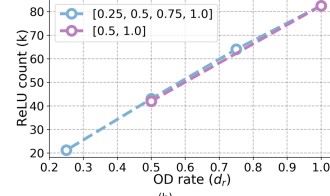 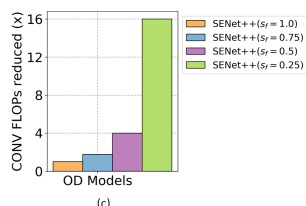

Figure 9: (a) Accuracy vs. OD rate, (b) ReLU count vs. OD rate for SENet++ training with different OD rate supports (*two* and *four*), (c) CONV layer FLOPs reduction factor for models at different OD rate values. We used ResNet18 on CIFAR-100 for this evaluation.

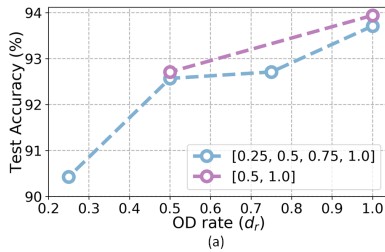 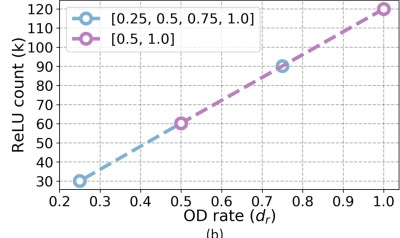

Figure 10: (a) Accuracy vs. OD rate, (b) ReLU count vs. OD rate for SENet++ training with different OD rate supports (*two* and *four*). We used VGG16 on CIFAR-10 for this evaluation.

random locations of the corresponding ReLU mask tensors. Table 9 clearly demonstraates the benefits of stage 2 as the model can provide improved performance of 11.28% compared to the one trained without stage 2.

Table 9: Importance of ReLU mask identification stage (stage 2). We used CIFAR-100 dataset.

| Model | Baseline Acc% | #ReLU (k) | ReLU mask identification (stage 2) | Test Acc% | Acc%/ #1k ReLU |
|---|---|---|---|---|---|
| | | 24.6 | ✗ | 59.12 | 2.37 |
| ResNet18 | 78.05 | 24.6 | ✓ | **70.59** | 2.87 |

**Ablation Studies for Stage 3 Importance.** Table 10 shows the importance of fine-tuning stage (stage 3). In particular, the accuracy difference of a model before and after stage 3 training can vary up to 6.3%.

Table 10: Importance of Activation similarity maximization (stage 3). We used CIFAR-100 dataset.

| Model | Baseline Acc% | #ReLU (k) | Activation similarity maximization (stage 3) | Test Acc% | Acc%/ #1k ReLU |
|---|---|---|---|---|---|
| | | 24.6 | ✗ | 64.10 | 2.57 |
| ResNet18 | 78.05 | 24.6 | ✓ | **70.59** | 2.87 |

**Ablation Studies with the PRAM Loss Component.** Table 11 shows the importance of PRAM loss component during the fine-tuning stage (stage 3). In particular, the accuracy can improve up to 0.69% on CIFAR-10 as evaluated with ResNet18 for 150k ReLU budget.

Table 11: Importance of PRAM loss at final fine-tuning stage. We used CIFAR-10 dataset.

| Model | Baseline Acc% | #ReLU (k) | With PRAM loss | Test Acc% | Acc%/ #1k ReLU |
|---|---|---|---|---|---|
| | | 150 | ✗ | 94.12 | 0.62 |
| ResNet18 | 95.2 | 150 | ✓ | **94.91** | 0.63 |

