# OpenReview forum: "Learning to Linearize Deep Neural Networks  for Secure and Efficient Private Inference"
_ICLR.cc/2023/Conference — ICLR 2023 poster_

### Official Review · Reviewer_e8en · 2022-10-20

**Confidence:** 3
**Correctness:** 3
**Technical Novelty And Significance:** 3
**Empirical Novelty And Significance:** 3
**Recommendation:** 8

**Clarity, Quality, Novelty And Reproducibility:**

The paper is well-written and clearly presented. The proposed method SENet++ is novel as one integrated framework. Reproducibility may be unstable due to the multi-stage nature of the proposed method.

**Strength And Weaknesses:**

Strengths:
1. A well-written paper with clear reasoning and structured presentation.
2. Proposes a novel linearization method for secure PI
3. The method seems to be extremely effective, compared to existing methods.
4. Extensive empirical evaluations and understandings.

Weaknesses:
1. Some of the techniques in SENet++ are adapted or directly transferred from existing works. For example, sensitivity evaluation
before training from Lee et al. (2018), ordered dropout from Horvath et al. (2021). This reduces part of the technical contribution of this paper.
2.  Missing an ablation study of the key components of SENet++. How much each component contributes to the final performance and f they can be integrated into existing methods to improve their performance.

**Summary Of The Paper:**

This paper presents a new SOTA method SENet++ for the reduction of ReLUs for Private Inference (PI). The method is based on a novel measure of ReLU sensitivity (of a layer), a learnable mask applied at each layer, a distillation technique, and ordered channel dropout, to remove the least important ReLU operations while maintaining high accuracy. Extensive experiments with comparisons to existing methods verify the superiority of the proposed method in terms of communication-saving and accuracy maintenance.

**Summary Of The Review:**

This paper presents a promising solution for ReLU reduction. The paper is well-written and the experiments are fairly thorough, though missing an important ablation study. The improvement of the proposed SENet++ over existing methods seems quite significant.

--------
Score has been raised to 8 after the rebual.

---

### Official Review · Reviewer_uQYb · 2022-10-25

**Confidence:** 3
**Clarity, Quality, Novelty And Reproducibility:** See above
**Correctness:** 4
**Technical Novelty And Significance:** 3
**Empirical Novelty And Significance:** 3
**Recommendation:** 6

**Strength And Weaknesses:**

Strength:
1. This paper studies a very important question.
2. They study the relation between ReLU importance and pruning sensitivity.
3. They propose a three-stage training process, SENet, to yield secure and efficient networks for PI  with the ReLU budget guarantee.
4. They propose SENet++ to further reduce both linear and ReLU layer compute cost,
5. They conduct extensive experiments on various models and show the improvements.
Weakness:
I am not an expert in this domain. The methodology and experiments sound reasonable to me. A minor suggestion is that since many models leverage the Vision Transformer as the backbone, I believe adding the experiments on ViT will be much more interesting.

**Summary Of The Paper:**

This paper studies how to reduce ReLU operation more efficiently to reduce the communication and latency overhead of privacy inference. They demonstrate the relation between a layer’s sensitivity towards pruning and its associated ReLU sensitivity and introduce an automated layer-wise ReLU sensitivity evaluation strategy.  They propose SENet, a three stage automated ReLU trimming strategy that can  yield models for a given reduced ReLU budget.

**Summary Of The Review:**

See above

---

### Official Review · Reviewer_oSdq · 2022-10-27

**Confidence:** 3
**Correctness:** 3
**Technical Novelty And Significance:** 3
**Empirical Novelty And Significance:** 3
**Recommendation:** 6

**Clarity, Quality, Novelty And Reproducibility:**

Clarity & Quality: This is one of my major points of criticism. While some sections like the introduction and related work are written clearly, later are rather hard to understand. For example, in section 3, the fourth paragraph. It is not clear to me how exactly the ReLUs were masked in the models, e.g. "We performed the same operation to evaluate the accuracy with ReLU only after the first CONV layer." Which conv layer is mentioned here, the first of the network or the first of the block?

Also, the presentation should be improved. Besides some typos and missing spaces, the size of the figures should be increased. For example, Figure 3 is hard to read without a large zoom. Also, some equations are overlapping with text, e.g., on the top part of page 5. Also, some variables such as r_final^l are not explicitly introduced in the text and their meaning must be inferred from the context.

I am unsure about the motivation provided in section 3. The inspiration is based on the results depicted in fig. 2 and the visualized inverse correlation between layer-wise pruning sensitivity and ReLU importance. However, I am not sure if the depicted correction is strong enough to motivate the approach. The ReLU sensitivity in the last four blocks ranges between 80% and 95%, which is a rather small range compared to the pruning sensitivity that decreases significantly after 7 and 10 layers.

Regarding the experimental section, it is hard to compare SENet to previous approaches, if the #ReLU parameters vary between the different approaches. Also, why are different models used to compare, for example, to the work of Cho et al (2022) and Jha et al. (2021) on top of table 5? One approach is evaluated on VGG16, the other on ResNet18, and with a different number of ReLU.

Also, ResNet-18 and 34 are no good choices for CIFAR since the models are adjusted to ImageNet-Scale (224x224) and perform strong pooling in the early layers. Using a ResNet adjusted to CIFAR, e.g., a ResNet-20, might have been a better choice. Take a look at the original ResNet Paper [He et al., Deep Residual Learning for Image Recognition], where the different architectures are explained.


Novelty: The approach seems to be novel, but I am not deep into the literature in the area of private inference. I acknowledge the novelty of the approach inspired by weight pruning.

Reproducibility: The paper provides the most important hyperparameters and source code. I did not run the code but expect the results to be reproducible.

Smaller Remarks:
- page 2, typo: "However, such approaches are extremely hyperparameter sensitive and often [do] not guarantee ..."
- page 3, paragraph "private inference": introduce abbreviation "SS"
- page 4: Move Figure 2 up to avoid the text being split in some sense.
- equation 3: A closing mark is missing.
- page 5, section 4.2: A space is missing after the first sentence
- table 2: The top should be aligned with the text

Further questions:
- Regarding the motivation based on the ReLU importance. If I understand this section correctly, all ReLUs are removed but the one from a specific ResNet block. So all other activation functions in the model are removed. Wouldn't those parts of the network simply collapse to a simple linear regression and could be represented by a single linear layer?
- Regarding the insight, that the ReLU importance increases in later layers: the number of channels in each ResNet block is doubled compared to the previous one, so the last blocks have a much higher number of channels and, therefore, parameters available. I could imagine this to be another reason why the prediction accuracy is better for the later blocks.

**Details Of Ethics Concerns:**

I do not have any ethical concerns.

**Strength And Weaknesses:**

Strength:
+ The inspiration based on weight pruning provides an interesting view of the topic.
+ Some experimental results support the improvements made by the proposed solution
+ The introduction and background and related work sections are well-written and motivated
+ Experiments were performed on various datasets and model architectures

Weaknesses:
- The presentation and writing of the paper should be improved (see more details in the quality section)
- Some parts of the paper are hard to follow, especially the second part of section 3 and parts of section 4.
- The experimental evaluation and comparison to previous approaches are not consistent. For example, the number of ReLUs allowed varies in Table 5 between SENet and other approaches. This makes it hard to assess the effectiveness of the approach.

**Summary Of The Paper:**

The paper proposes a new approach to reduce the number of ReLU activation functions in a given model. In private inference applications, latency plays a crucial role and is highly dependent on the number of ReLUs. The proposed approach is based on the insight that parameter pruning sensitivity and ReLU importance are negatively correlated. It then uses a three-step algorithm to select a ReLU mask and distill the knowledge from a teacher model. The paper further proposes an additional improvement to not only reduce the number of activation functions but also the overall model size by leveraging ordered dropout.


**Summary Of The Review:**

I like the motivation taken by parameter pruning and moved to "ReLU pruning". This offers a novel perspective on the problem of nonlinearities in the context of efficient private inference. However, the paper lacks clarity and quality of writing. Also, the comparison to previous approaches seems inconsistent and makes it hard to evaluate the overall effectiveness of the proposed solution.

---

### Decision · Program_Chairs · 2023-01-20

**Decision:**

Accept: poster

**Justification For Why Not Higher Score:**

Technically the paper is reasonable. It lacks clarity and accessibility  (especially for people not working precisely on this topic), and demands more proofreading.

**Justification For Why Not Lower Score:**

This paper offers a novel perspective on the problem of nonlinearities in the context of efficient private inference, the results are great and the reviewers are unanimously positive.

**Metareview: Summary, Strengths And Weaknesses:**

This paper studies how to reduce ReLU operation more efficiently to reduce privacy inference's communication and latency overhead. The authors rely their algorithm on the insight that parameter pruning sensitivity and ReLU importance are negatively correlated. They then propose a three-step algorithm to select a ReLU mask and distill the knowledge from a teacher model. Additionally, they study how to reduce overall model size besides only reducing the number of activation functions.

The reviewers identified the main strengths and weaknesses as follows:
+ (S) This paper offers a novel perspective on the problem of nonlinearities in the context of efficient private inference
+ (S) The novel linearization method looks pretty effective compared to its counterparts
+ (S) Experiments report on various datasets and model architectures
- (W) Writing quality needs improvement (improved during rebuttal revision)
- (W) Some ablation studies of SE-Net++ were missed

After rebuttal, the authors address the raised concerns to satisfaction, and all reviewers reach a positive consensus.

**Note From Pc:**

if the above contains the word "oral" or "spotlight" please see: "oral" presentation means -> notable-top-5% and "spotlight" means -> notable-top-25%. As stated in our emails, we are disassociating presentation type from AC recommendations